# Effects of Combined Exercise Program on Spinal Curvature and Balance Ability in Adolescents with Kyphosis

**DOI:** 10.3390/children9121999

**Published:** 2022-12-19

**Authors:** Yun-Jin Park, Won-Moon Kim, Jae-Ho Yu, Hyung-Hoon Moon, Yong-Gon Seo

**Affiliations:** 1Division of Health Rehabilitation, Osan University, Osan-si 18119, Gyeonggi-do, Republic of Korea; 2Department of Sports Science, Dongguk University, Gyeongju-si 38066, Gyeongsangbuk-do, Republic of Korea; 3Department of Physical Therapy, Sunmoon University, Asan-si 31460, Chungcheongnam-do, Republic of Korea; 4Department of Sports Medicine, Cha University, Pocheon-Si 11160, Gyeonggi-do, Republic of Korea; 5Division of Sports Medicine, Department of Orthopedic Surgery, Samsung Medical Center, Seoul 06351, Republic of Korea

**Keywords:** postural kyphosis, complex exercise, Cobb’s angle, balance, spinal curvature

## Abstract

Thoracic hyperkyphosis is associated with postural abnormality, physical function, and quality of life. This study aimed to examine the effects of a combined exercise program on the spinal curvature and balance ability in adolescents with kyphosis. Fifty-one adolescents (mean age 21.95 ± 3.90 year, 23 male and 28 female) diagnosed with kyphosis were randomly divided into two groups: an experimental group (n = 25) and a control group (n = 26). All participants in the study group underwent a combined exercise program for 60 min, three times per week, for 12 weeks. Cobb’s angle and forward head angle showed significant differences between the two groups (*p* < 0.001). The anterior and posterior weight distributions of static and dynamic plantar foot pressures showed a significant difference between the two groups (*p* < 0.001), and significant differences were observed only in the study group (*p* < 0.001). However, the left and right static plantar foot pressures did not differ significantly. In conclusion, these results demonstrate that a combined exercise program is an effective intervention for the improvement of alignment in the spinal curve and balance in adolescents with postural kyphosis.

## 1. Introduction

Postural kyphosis is a disease that often starts in adolescence during a period of growth and results in a continuous increase in kyphosis with increasing age [1], which is caused by complex factors such as incorrect posture and habits [2]. In recent years, the implementation of online classes due to COVID-19 has caused a rapid increase in the use of smartphones and PCs, leading to adolescents adopting inappropriate postures, which increases the structural deformation of the spine [3].

Adolescence is a period of growth where physical development is not fully achieved; therefore, an increase in kyphosis of the spine causes muscle weakness and imbalance, neck and back pain, and decreased joint range of motion, thereby reducing the ability to balance the body [4]. The ability to control balance is a major factor in the human ability to perform all functions. To control balance, it is necessary to exert control through the nerve root and integrate information from the visual, somatosensory, and vestibular systems [5]. To maintain balance, it is important to keep the center of gravity within the human body and the center of weight within the support base and consider the information coming from the feet in contact with the ground at the time of posture adjustment [6].

Kyphosis moves the human body’s center of gravity backward, resulting in an inability to support the vertical load on the trunk and spine, making the thoracic spine excessively curved [7]. The degree of spinal deformity directly influences foot weight distribution [8]. Additionally, along with the deformation of the spine and change in the human body’s center of gravity, lordosis of the cervical spine also increases, compensating where balance function is reduced due to limited vision [9].

Plantar foot pressure is an index of the qualitative status of gait and balance. To test foot pressure, it is possible to directly observe the characteristics of the sole supporting the weight, and necessary to compare and analyze the pressure applied to the sole during specific movements [10]. Methods for improving postural kyphosis include spinal fusion, fixation, and conservative therapy. Combined exercises, including stretching, corrective exercises, and strengthening exercises, are mainly applied in clinical settings [11,12]. According to previous studies, exercise improves kyphosis; however, studies regarding the effects of complex exercises on postural kyphosis and balance ability in adolescents are lacking. Therefore, this study aimed to investigate the effects of combined exercise on Cobb’s angle, forward head angle (FHA), and balance ability in adolescents with postural kyphosis.

## 2. Materials and Methods

### 2.1. Participants

This study included 60 adolescents between the ages of 10 and 19 years who were diagnosed with postural kyphosis (Cobb’s angle: 30–45°) on an X-ray of the whole spine anterior–posterior (AP) by an orthopedic surgeon among patients who visited *p* orthopedics in Seoul, Korea. Nine of the sixty subjects dropped out of this study due to the following reasons: five subjects refused to participate in this study and four had experience performing exercise intervention. Finally, 51 participants were randomly divided into two groups: an exercise group with 25 participants and a control group with 26 participants. The inclusion criteria were as follows: (1) those who had not undergone conservative treatment for kyphosis in the past and had no orthopedic abnormalities other than kyphosis, (2) those who did not have any mental or neurological abnormalities, and (3) those who did not have motor ability abnormalities such as balance ability and visual impairment. Those who received exercise intervention for hyperkyphosis or who had abnormal neurological function or who had spinal disc disease were excluded for this study. The physical characteristics of the participants are listed in Table 1. All subjects received sufficient explanations regarding this study and signed an informed consent form before participation in this study. This study was approved by the Research Bioethics Committee of Dongguk University (DGU IRB No. 20220018, approval date: 17 August 2022).

### 2.2. Study Design

All participants received pre-intervention examinations; the following characteristics were measured: height, weight, Cobb’s angle using X-ray, FHA using the photographic method, and balance ability using plantar foot pressure. After the examination, all participants were randomly divided into the two groups mentioned previously. In this study, the study group performed combined exercises for 60 min, three times a week, for 12 weeks. The control group did not perform the combined exercise program and they were instructed to continue their lifestyle. The post-test examinations were performed in the same manner as the pre-test examinations after 12 weeks for both groups.

### 2.3. X-Ray Examination to Measure Cobb’s Angle and Forward Head Angle

During the X-ray examination, the participant flexed both upper extremities forward to 90° in an upright position and an X-ray was performed to examine the side of the spine from the head to the hips. Cobb’s method was applied to measure the Cobb’s angle; a line was drawn vertically from the upper epiphyseal plate of the fifth thoracic spine and another line was drawn from the lower epiphyseal plate of the twelfth thoracic spine to obtain the intersecting angle (Figure 1). In order to measure forward head angle, GPA (alFOOTs, Seoul, Republic of Korea) was used in this study. Landmarks were attached to the right ear tragus and the seventh cervical vertebra in an upright position. After the landmarks, the subject bent forward three times and reached overhead three times, and the subjects were instructed to adopt a comfortable and upright posture, gazing to the front [13]. Photographs were taken three times within 5 s, and the results were analyzed by the PAS (alFOOTs, Seoul, Republic of Korea) program. The angle was measured between the two lines; one is vertical anteriorly and other is a line connecting the tragus and the C7 marker. The average value of the three images was used for the data analysis [11] (Figure 2).

### 2.4. Static and Dynamic Plantar Pressure

The Gaitview AFA-50 system (alFOOTs, Seoul, Republic of Korea) was used to examine static plantar foot pressure [14]. First, participants were asked to stand in a comfortable position on the Gaitview system with both hands placed on the navel area and to hold the same posture for 15 s. The static plantar foot pressure was analyzed by dividing the sole into the front and rear sections and left and right sections according to the pressure distribution of the sole, and the average value was used. For the dynamic plantar pressure measurement, the process of taking one step at a time during gait was examined. The measured value was obtained using the sum of the pressure ratio of the forefoot and the total pressure ratio. The same method was used for the measurements of the rear foot.

### 2.5. Combined Exercise Program

The center-based exercise program was performed with an exercise specialist for 60 min per day, three times a week, for 12 weeks in the order of warm-up, main exercise, and cool-down. Regarding the combined exercise program, stretching of major joints and muscle groups using small tools, neck exercises using bands, trunk exercises using body weight, and Schroth method exercises for improving muscle elongation and muscle strength were performed [15,16]. The Borg rating of perceived exertion (RPE) 6–20 scale was used for exercise intensity; warm-up and cool-down were set to RPE 11–13; and the main exercise was initiated at RPE 14–16 and gradually increased. The rest period between types of exercises was set to 60 s, with 30 s between sets. Details of the program are listed in Table 2.

### 2.6. Data Analysis

Analysis of descriptive statistics was performed to compare baseline data, and the result was described as mean (M) and standard deviation (SD). An independent *t*-test was used to compare and analyze the changes between the two groups. The statistical program SPSS 22.0 for Windows (Armonk, NY: IBM Corp.) was used to analyze the data in this study. Statistical significance was set at *p* < 0.05.

## 3. Results

Cobb’s angle and FHA showed significant differences between the two groups (*p* < 0.001) (Table 3). Regarding static balance function, anterior and posterior weight distributions showed significant differences between the groups (*p* < 0.001). However, the left and right weight distributions did not show significant differences between the two groups (*p* = 0.120, *p* = 0.451, respectively). Regarding dynamic balance function, anterior and posterior weight distributions showed significant differences between the two groups (*p* < 0.05) (Table 4).

## 4. Discussion

This study evaluated changes in Cobb’s angle, FHA, and balance ability after applying a combined exercises program in adolescents with thoracic hyperkyphosis. The results of this study showed that Cobb’s angle and FHA showed significant improvements in the study group.

Hyperkyphosis of the thoracic spine causes weakness of the cervical flexors and abdominal muscles and increases lordosis of the cervical spine due to the shortening of the spinal extensors and pectoral muscles [17]. This moves the center of mass of the head forward [18], and to compensate for this, it is necessary to increase the tension, shorten the cervical extensors, and weaken the deep head and neck flexors [7].

In this study, FHA showed a significant decrease after applying a combined exercise program in the exercise group. A study conducted by Seidi et al. [11] reported results similar to ours. The study suggests that the reason why the cervical vertebra position angle showed a significant improvement was due to the application of strengthening and flexibility exercises for the neck retraction and flexor muscles to reduce the angle of lordosis of the cervical vertebrae. Therefore, the results of this study could also be explained with the same mechanism. Further study is needed to explain, in more detail, the mechanism related to the decrease in cervical position angle.

Cobb’s angle is the most widely used measurement to determine the magnitude of spinal deformities and the method was used in the study group. The result of Cobb’s angle in this study showed a significant decrease after intervention. A previous study conducted by Daniel et al., (2007) [16] reported that therapeutic complex exercise is associated with a reduction in thoracic kyphosis. The Schroth exercise (applied thoracic expansion through breathing while providing feedback so that the person can accurately recognize inappropriate posture alignment) is a beneficial intervention to improve Cobb’s angle in patients with adolescent idiopathic scoliosis [15]. Therefore, the results of this study could be summarized by stating that the combined exercise, including stretching, strengthening, and the Schorth exercise, affects correction of muscle imbalance and kyphosis by increasing recognition of appropriate postural alignment.

The ability to balance in the sagittal plane is reduced due to the center of gravity being moved to the rear with a gradual increase in kyphosis of the spine, resulting in the function of balance decreasing by changing the body’s center [9]. For comparison of static and dynamic balance, the pressure distribution on the sole was used in this study. The study results revealed that the anterior and posterior weight distributions of static and dynamic plantar foot pressure were significantly different between the two groups. Analysis of the pressure area for the weight distribution in both the static and dynamic plantar foot pressure tests indicated that the posterior pressure area decreased and the anterior pressure area increased. We thought that the combined exercise program had an important role in moving the position of the center of gravity from the rear to the front, and that it could reduce the kyphosis angle and the position angle of the cervical vertebrae. In normal gait, the path of plantar foot pressure in the stance phase starts from the posterior and lateral sides of the posterior heel, passes through the medial midfoot, passes through the second metatarsal head, passes between the big toe and second toe, and ends at the anterior and medial sites of the foot. Thus, it is important to evaluate the equilibrium sense or gait pattern [19]. Accordingly, the increase in the anterior area of dynamic plantar foot pressure in this study indicates that it is possible to walk with increased stability when switching from the swing phase to the stance phase during gait [20].

This study has some limitations. Firstly, this study did not comment on which exercises had more influence on the results among the applied exercise interventions. Further study is necessary to compare the effects between applied exercises. Secondly, pelvic movement is associated with posture in thoracic kyphosis, but this study did not measure variables related on pelvic movement. Thirdly, strengthening exercises were included, but the measurement of back muscle strength was not performed in the present study. An additional study is required to confirm the effects of muscle strength on the change in thoracic angle.

## 5. Conclusion

This study showed that a combined exercise program is an effective intervention for improvement in the curvature of the spine in adolescents, thereby helping improve static and dynamic balance, which is a meaningful result. Further studies are needed to determine the effect of pelvic displacement according to the degree of spine curvature deformation. Furthermore, a segmented exercise program that considers the degree of deformation of the curvature of the cervical and lumbar spine due to kyphosis and posture habits should be developed.

## Figures and Tables

**Figure 1 children-09-01999-f001:**
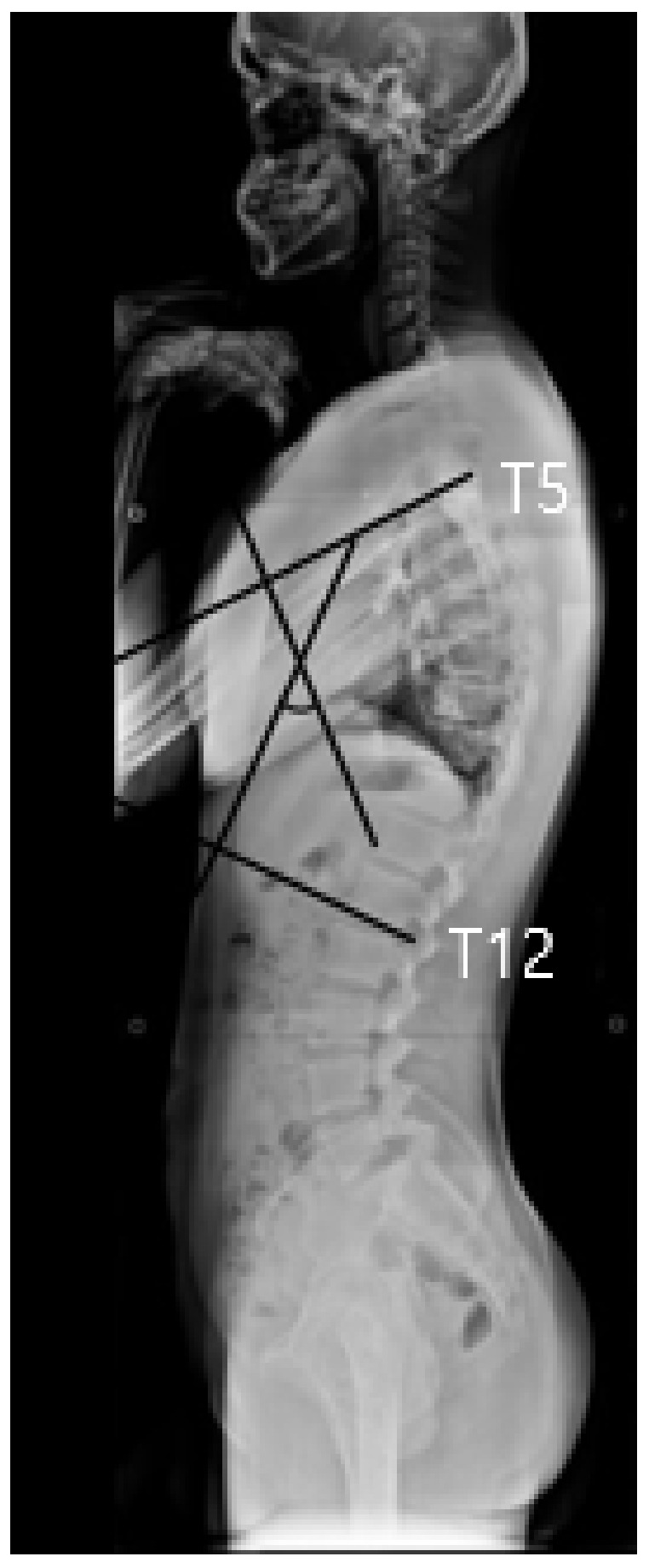
Thoracic kyphosis measured from the upper endplate of the T5 to the lower endplate of the T12 using Cobb’s method.

**Figure 2 children-09-01999-f002:**
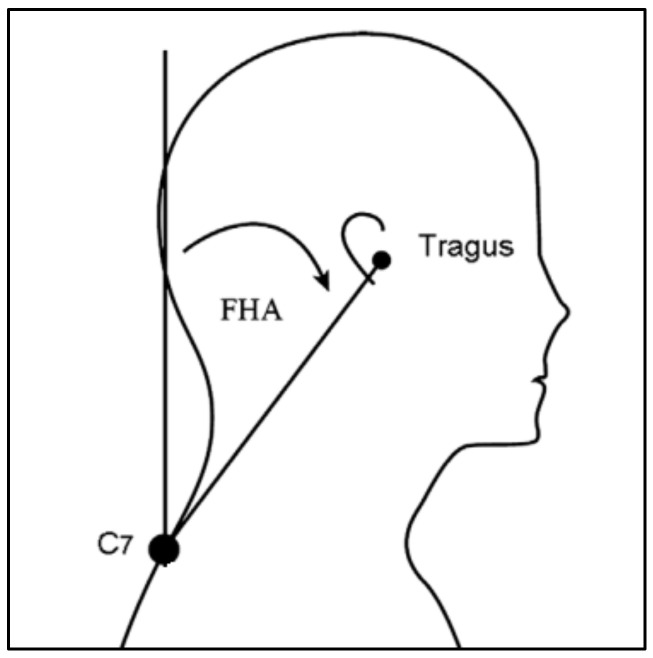
Forward head angle measured from the vertical anteriorly to a line connecting the tragus and the C7 marker.

**Table 1 children-09-01999-t001:** Physical characteristics of the participants.

Variables	Exercise Group (n = 25)	Control Group (n = 26)	*p*-Value
Age (year)	17.35 ± 1.23	17.72 ± 2.13	0.30
Height (cm)	167.61 ± 7.06	168.93 ± 7.00	0.87
Weight (kg)	58.97 ± 9.17	61.36 ± 10.63	0.26
BMI (kg/m^2^)	20.89 ± 2.03	21.39 ± 2.16	0.447
Sex (Male/Female)	9/16	14/12	-
FHA (°)	16.23 ± 1.81	14.88 ± 2.33	0.34
Cobb’s angle (°)	38.22 ± 3.76	35.00 ± 4.08	0.78

Values are presented as mean ± standard deviation. FHA, Forward head angle.

**Table 2 children-09-01999-t002:** Combined exercise program.

Division	Workout types	Frequency	RPE
Warming up	Treadmill: gait corrective program	5 min	11–13
Combined exercise program	Stretching of major joint and muscle group – suboccipitalis stretching – latissimus dorsi and pectoralis stretching – roll form and gym ball stretching Band Exercise – chin—in exercise – neck retraction and flexion exercise Back Muscle Strengthening – trunk extension exercise – thoracolumbar mobilization exercise – prone arm and leg raise exercise – scapula adduction exercise Schroth Method Exercise – sitting question mark – standing in a doorway – kneeling between chairs	Each exercise 3 sets 1 set/20 reps (50 min)/ each exercise 60 s rest	14–16
Cool down	Treadmill: gait corrective program	5 min	11–13

RPE, rating of perceived exertion.

**Table 3 children-09-01999-t003:** Result of changes in structural variables.

Variable	Group	Pre-Intervention	Post-Intervention	*p*-Value
FHA (°)	EG	16.23 ± 1.81	10.77 ± 3.35	0.000 ***
	CG	14.88 ± 2.33	15.56 ± 6.11	
Cobb’s angle (°)	EG	38.22 ± 3.76	26.95 ± 5.03	0.000 ***
	CG	34.12 ± 2.90	33.33 ± 2.72	

Values are presented as the mean ± standard deviation. FHA, Forward head angle; EG, exercise group; CG, control group. Significant differences were determined using independent *t*-tests. ***: *p* < 0.001.

**Table 4 children-09-01999-t004:** Result of changes in structural variables.

	Variable		Group	Pre	Post	*p*-Value
Static (kPa)	Anterior		EG	33.42 ± 7.69	48.50 ± 6.77	0.000 ***
			CG	39.03 ± 7.97	40.08 ± 7.03	
	Posterior		EG	66.58 ± 7.69	51.90 ± 6.20	0.000 ***
			CG	60.59 ± 8.21	59.54 ± 7.11	
	Left		EG	51.13 ± 7.58	50.67 ± 2.21	0.120
			CG	48.50 ± 6.39	48.43 ± 6.85	
	Right		EG	68.48 ± 101.46	49.32 ± 2.22	0.451
			CG	50.26 ± 7.98	50.42 ± 6.98	
Dynamic (kPa)	Left	anterior	EG	20.38 ± 7.42	25.75 ± 4.20	0.015 *
			CG	22.18 ± 5.76	22.38 ± 5.27	
		Posterior	EG	30.32 ± 9.29	24.29 ± 3.63	0.020 *
			CG	28.55 ± 6.72	28.16 ± 7.31	
	Right	anterior	EG	19.92 ± 6.59	25.27 ± 3.33	0.040 *
			CG	22.56 ± 4.90	23.10 ± 4.00	
		Posterior	EG	29.40 ± 7.72	24.70 ± 4.13	0.044 *
			CG	26.96 ± 5.40	27.29 ± 4.82	

Values are presented as the mean ± standard deviation. EG, exercise group; CG, control group. Significant differences were determined using independent *t*-tests. *: *p* < 0.05; ***: *p* < 0.00.

## Data Availability

The data used to support the findings of this study are available from the corresponding author upon reasonable request.

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
