# Peer review of "Effects of Combined Exercise Program on Spinal Curvature and Balance Ability in Adolescents with Kyphosis"

_children, 2022, doi:10.3390/children9121999_

Round 1

Reviewer 1 Report

This study asks an interesting question, however, the study design presents several difficulties:

1. Duration and specific exercises were prescribed. How was compliance with the excercise program measured? Self-reported, PT reported, use of an App, use of an activity monitor? Was improvement in outcomes measures "dose" dependent?

2. An additional control group would be very interesting and useful. These individuals would be "instructed" on how to stand for the 12 week post-treatment xray but they would not do any excercises over the course of the study. It is entirely possible the significant xray changes in the excercise group are secondary to "coaching" and specific direction from the PT/MDs as to desired posture and not due to actual completion/performance of specific excercises.

3. The photographic method for determination of Forward Head Angle was performed 3 times. Was this method reliable? Is it valid? If so, please list references and reliability. Why was the measurement done 3 times? Were patients instructed to look at a specific point in front of them for this analysis? How is the FHA affected by point of eye focus? Is it possible to maintain forward horizontal gaze while tilting the head back and looking slightly down?  

4. Is the GaitView system a validated methodology for measurement of balance? If so, please give references.

Author Response

This study asks an interesting question, however, the study design presents several difficulties:

Point 1. Duration and specific exercises were prescribed. How was compliance with the exercise program measured? Self-reported, PT reported, use of an App, use of an activity monitor? Was improvement in outcomes measures "dose" dependent?

Response: Thank you for your good indication. The exercise program is based on the center-based intervention during the period of this study. We add the sentence in combined exercise program section. The compliance to exercise was confirmed by the subject with self-reported. Based on the result of the exercise intervention, we think that exercise dose is associated with improvement of the outcome.

Point 2. An additional control group would be very interesting and useful. These individuals would be "instructed" on how to stand for the 12 week post-treatment x-ray but they would not do any exercises over the course of the study. It is entirely possible the significant x-ray changes in the exercise group are secondary to "coaching" and specific direction from the PT/MDs as to desired posture and not due to actual completion/performance of specific exercises.

Response: The authors agree with your opinion. In this study, the control group did not receive any exercise intervention and the coaching may be affected the result of X-ray change. Although we described that the muscle strength did not measure in limitation section, our study needed to analyze the effect of the muscle strength on X-ray change for verifying the effect of combined exercise program. Thanks again for the good point.

Point 3. The photographic method for determination of Forward Head Angle was performed 3 times. Was this method reliable? Is it valid? If so, please list references and reliability. Why was the measurement done 3 times? Were patients instructed to look at a specific point in front of them for this analysis? How is the FHA affected by point of eye focus? Is it possible to maintain forward horizontal gaze while tilting the head back and looking slightly down?

Response: First of all, we apologize for giving the confusion caused by not describing the study method in detail. The study method on the analysis of forward head angle was conducted based on a previous study, which was cited as reference number of 13, and the photographic method have reported with reliability. We inserted the reference to the method section. In this study, the 3 times were measured within 5 seconds to increase the reliability according to the study method of previous study. Also, we add more detail information on the study method in our manuscript for the reader. We list on references related to reliability of a photographic method in the below. Thank you for your good indication.

References

Pausic J, et al. Reliability of a photographic method for assessing standing posture of elementary school students. J Manipulative Physiol Ther. 2010, 33, 425-31.

Hazar Z, et al. Reliability of photographic posture analysis of adolescents. J Phys Ther Sci, 2015, 27, 3123-3126.

Point 4. Is the GaitView system a validated methodology for measurement of balance? If so, please give references.

Response: We think the validity of the equipment for the test is very important. Several studies have been conducted using the GaitView system and one study reported on the reliability of this equipment. Therefore, we suggest several studies using GaitView system and inserted the reference “Kim YT, Lee JS, Normal pressure and reliability of the GaitView® system in healthy adults. Prosthetics and orthotics international. 2012;36:157-64.” to the method section. Thanks again.

Reference

Bae KH, et al. Analyses of plantar foot pressure and static balance according to the types of insole in the elderly. Korean journal of sport biomechanics. 2016;26:115-126.

Lee JS, et al. Correlation of foot posture index with plantar pressure and radiographic measurements in pediatric flatfoot. Annals of rehabilitation medicine.2015;39:10-17

Reviewer 2 Report

Title: Effects of combined exercise program on spinal curvature and balance ability in adolescents with kyphosis

 Article Type: Article

 Summary

 In this study, the authors aimed to examine the effects of a combined exercise program on the spinal curvature and balance ability in adolescents with kyphosis. For this purpose, 51 participants diagnosed with kyphosis were recruited and randomly divided into two groups (intervention (a combined exercise program) and control). The results indicated that cobb's angle and forward head angle as well as the anterior and posterior weight distributions of static and dynamic plantar foot pressures were different between two groups. As a conclusion, the authors reported that a combined exercise program is an effective intervention for the improvement of alignment in the spinal curve and balance in adolescents with postural kyphosis.

 Evaluation

 The topic of this study is interesting for publication in the Journal, but there are some questions that the author should solve and clarify those and the manuscript should be re-evaluated. Therefore, there are some points should be addressed by the authors, in order to improve the quality of the manuscript. 

 points

-          Please add mean age and SD to the abstract. Also add gender of the participants too.

-          Please add a related background to the first line of the abstract.

-          L 20, please change “a study group” to an experimental group.

-          How is the sample size calculated? Why you had only 51 participants? please describe this in the method section in detail.

-          Please add exclusion criteria to the method section.

-          What was your research method? please describe this in the method section.

-          What was your participant’s gender?

-          Please add participant’s BMI to table 1.

-          Please describe more about control group.

Author Response

Title: Effects of combined exercise program on spinal curvature and balance ability in adolescents with kyphosis

Article Type: Article

Summary

In this study, the authors aimed to examine the effects of a combined exercise program on the spinal curvature and balance ability in adolescents with kyphosis. For this purpose, 51 participants diagnosed with kyphosis were recruited and randomly divided into two groups (intervention (a combined exercise program) and control). The results indicated that Cobb's angle and forward head angle as well as the anterior and posterior weight distributions of static and dynamic plantar foot pressures were different between two groups. As a conclusion, the authors reported that a combined exercise program is an effective intervention for the improvement of alignment in the spinal curve and balance in adolescents with postural kyphosis.

Evaluation

The topic of this study is interesting for publication in the Journal, but there are some questions that the author should solve and clarify those and the manuscript should be re-evaluated. Therefore, there are some points should be addressed by the authors, in order to improve the quality of the manuscript.

Point 1. Please add mean age and SD to the abstract. Also add gender of the participants too.

Response: According to your comment, we add mean age±SD and gender to the abstract section.

Point 2. Please add a related background to the first line of the abstract.

Response: According to your comment, we add a related background to the first line of the abstract.

Point 3. L 20, please change “a study group” to an experimental group.

Response: According to your opinion, we modified “a study group” to an experimental group.

Point 4. How is the sample size calculated? Why you had only 51 participants? please describe this in the method section in detail.

Response: Thank you for making a very important point. We did not calculate the sample size using the program such as G*Power. We know that our sample size is small if it is calculated by G*Power program. The recruitment of subjects was very difficult in study period due to COVID-19. However, we tried to recruit similar sample size of 60 subjects for our study because previous study (reference 13 numbers) conducted with 60 participants and reported the effect of exercise intervention on thoracic hyperkyphosis. We described about sample size in the method section. The authors also think that future study should conduct the results of the study with large sample size. Thanks again for the good point.

Point 5. Please add exclusion criteria to the method section.

Response: According to your comment, we add exclusion criteria to the method section.

Point 6. What was your research method? Please describe this in the method section.

Response: Thank you for your good indication.

Point 7. What was your participant’s gender?

Response: We described the gender in table 1.

Point 8. Please add participant’s BMI to table 1.

Response: According to your comment, we add BMI to table 1.

Point 9. Please describe more about control group.

Response: Thank you for your comment. We also think that the information of control group is needed for reader. Therefore, the information was described in study design section.

Round 2

Reviewer 1 Report

Much improved manuscript